# Correlates of emotional violence against children in Rwanda: Findings from a cross-sectional national survey

**Alypio Nyandwi**[1]*, **Cyprien Munyanshongore**[1], **Laetitia Nyirazinyoye**[1], **Prata Ndola**[2], **Gisela Perren-Klingler**[3]

**1** University of Rwanda, Kigali, Rwanda, **2** School of Public Health, University of California, Berkeley, Berkeley, California, United States of America, **3** IPTS, Bale, Switzerland

* nalypio@gmail.com

## Abstract

### Introduction

National data on children affected by violence are critical in preventing violence against children. Rwanda conducted its first cross-sectional national survey on violence against children in 2015. This study used data from the Rwanda Survey to describe the profile of children affected by emotional violence (EV) and to assess factors associated with it in Rwanda.

### Methods

A sample of 1,110 children (618 boys and 492 girls) aged 13–17 from the Rwanda Survey was analysed. Weighted descriptive statistics were applied to describe the prevalence of EV and the profile of children affected by it. In addition, factors associated with EV were investigated using logistic regression.

### Results

Male children were more likely to experience EV than female children. Nine percent (8.87%, 95% CI [6.95–11.25]) of male children versus five percent (5.17%, 95% CI [3.79–7.03]) of female children reported having experienced EV in their lifetime. Seven percent (6.77%, 95% CI [5.15–8.84]) of male children versus four percent of female children (3.97%, 95% CI [2.83–5.54]) reported having experienced EV in the last twelve months before the survey. Fathers and mothers were the top two perpetrators of EV against children. Seventeen percent of male children (17.09%, 95% CI [11.06–25.47]) and 12 percent of female children (11.89%, 95% CI [6.97,19.55]) reported EV by their fathers. Mothers were responsible for nineteen percent (19.25%, 95% CI [12.94–27.65]) of EV reported by male children and eleven percent (10.78%, 95% CI [5.77–19.25]) of EV reported by female children. Female children (OR = 0.48, 95% CI [0.31–0.76]) and children with some trust in people from their communities (OR = 0.47, 95% CI [0.23–0.93]) were less likely to report EV. Factors associated with risk for EV were not attending school (OR = 1.80, 95% CI [1.10–2.92]), living with

**Data Availability Statement:** The dataset used to perform analyses presented in this manuscript is owned by a third party: the Rwanda Biomedical

Center. Therefore, anyone who needs to use it requests it to the Rwanda Biomedical Center (email: info@rbc.gov.rw) or the Principal Investigator (Yvonne Kayiteshonga, email: yvonne. kayiteshonga@rbc.gov.rw). The authors confirm that others would be able to access these data in the same manner as the authors.

**Funding:** The author(s) received no specific funding for this work.

**Competing interests:** The authors have declared that no competing interests exist.

father only (OR = 2.96, 95% CI [1.21–7.85]), not feeling close to biological parents (OR = 7.18, 95% CI [2.12–24.37]), living in a larger household (OR = 1.81, 95% CI [1.03–3.19]), not having a friend (OR = 2.08, 95% CI [1.02–4.11]), and not feeling safe in the community (OR = 2.56, 95% CI [1.03–6.38]).

## Conclusion

EV against children was pervasive in Rwanda, with parents topping the list of its perpetrators. Children from unsupportive socioeconomic family environments, i.e., children without a close relationship with biological parents, children not attending school, children living with their fathers only, children from larger households of five people and more, children without a friend, and children who reported not feeling safe in their communities, were identified as groups of children vulnerable to emotional violence in Rwanda. A family-centred approach, focusing on positive parenting and protecting vulnerable children, is needed to reduce emotional violence against children and the risk factors associated with it in Rwanda.

## Introduction

Emotional violence (EV) against children has damaging short- and long-term consequences on the child's mental, spiritual, moral, and socioeconomic development [1–5]. Globally, every year, one in three children is a victim of EV [6]. However, for some child protection experts, the global proportion of children suffering from EV would be underestimated because it is one form of child abuse that is hard to detect [7, 8]. EV is usually committed through nonverbal and verbal actions encompassing repeated bullying, humiliation, harassment, embarrassment, social isolation, verbal assaults, insults, threats, movement restrictions, neglect [9, 10], etc. These psychologically harmful practices do not leave physical traces of injuries easily identifiable on the victims to call for immediate attention on the side of concerned child protection professionals, welfare or criminal systems [8, 11, 12]. Quite often, the identification of EV largely depends on the disclosure by the victim through his narrated experiences [12], which is one of the reasons EV would be underestimated or given less attention than physical and sexual abuse [7, 8, 13].

Additionally, the underestimation of EV in low-income countries is amplified by limited technical and financial capacities to collect routine and survey data on child maltreatment [14]. In Africa, for example, many countries do not yet have national data on the burden of violence against children. By 2022, only fourteen African countries (Rwanda, Uganda, Kenya, Tanzania, Zambia, Lesotho, Malawi, Nigeria, Ivory-coast, Zimbabwe, Swaziland, Mozambique, and Botswana) had managed to conduct at least one round of a national survey on violence against children [15]. In these African countries that conducted national surveys on violence against children, the lowest prevalence of EV by a parent, caregiver or adult relative in the past 12 months among children aged 13–17 was reported in Lesotho, with 6.9% of EV in female children and 3.8% of EV in male children [16]. Conversely, the highest prevalence was reported in Uganda: 22.2% in girls and 22.8% in boys [17]. These findings fall in the range of results from a study that estimated the global prevalence of past-year violence against children using surveys on violence against children in 2016. In Africa, the prevalence of past-year EV against children aged 13–17 ranged from ten to over fifty percent [18].

For the abovementioned African countries, generating national data on the burden of violence against children is a commendable starting point. However, many more steps must be

taken to understand each country's context of risk factors for violence against children. Some of the steps that can be envisaged include repeating the same surveys regularly and scrutinising data collected through national surveys to have a broader picture of patterns of the prevalence of violence against children and to inform the design and implementation of preventive interventions. On the other hand, routine data from settings and systems dealing with children should supplement survey data to create and maintain an active identification of cases, causes, and risk factors for violence against children [19]. Specifically, there should be special attention to EV in Africa because, in the context of limited data on violence against children, it remains the least studied child maltreatment compared to physical and sexual violence in children. Furthermore, very few studies have focused on emotional child abuse alone, and the risk and protective factors of EV in sub–Saharan Africa are less documented. In available small-scale data, EV and its associated factors are studied together with other adverse childhood experiences [20, 21].

For Rwanda, one of the African countries that has conducted a national survey on violence against children, the national prevalence of past year EV was 8% in female children and 13% in male children aged 13–17. As in many other countries that have conducted national surveys on violence against children in Africa, the Rwanda survey report did not present factors associated with violence against children, including EV. To fill that gap, this study analysed data from the Rwanda survey on violence against children to respond to two questions:

1. What are the background characteristics of children who reported EV in Rwanda?

2. What are the correlates of EV against children in Rwanda?

This study aimed to go beyond the prevalence of EV presented in the Rwanda Violence Against Children and Youth Survey Report and describe the profile of children who experience EV in Rwanda. Information on children who are more likely to be victims of EV would inform the design and implementation of well-targeted interventions for preventing EV in children. In the same way, the identification of factors associated with EV in children would help EV preventive interventions to be focused on real issues.

## Methods

The analyses presented in this study are based on data from the violence against children and youth survey (VACS) conducted in Rwanda in 2015. The VACS is a standardized survey developed by the Centers for Disease Control and Prevention (CDC) to measure physical, emotional, and sexual violence against children and youth up to age 24 [22]. In addition, the national household survey captures information on sexual, emotional and physical violence, perpetrator types, event location, health outcomes, risk behaviours and protective factors [11, 23]. The CDC technically supports the design and implementation of VACS in countries.

## Rwanda violence against children and youth survey

### Survey participants

Eligible participants were females and males aged 13–24 who could speak Kinyarwanda or English in a sampled household in Rwanda. Children and youth with mental disabilities, with limited capacity to understand the questions being asked, and those with hearing and speech impairment that prevented them from participating in an oral interview for the survey were excluded from the study. However, to fill that gap in children and youth with disability, a separate qualitative violence survey was conducted on children and youth with disability in care institutions in Rwanda [24].

## Sample size

A total sample of 1,180 males and 1,032 females participated in the Rwanda VACS. The overall response rate was 98% for males and 97% for females [23, 25]. Rwanda VACS applied a three-stage split sample design to obtain the total sample. In the first stage, 250 enumeration areas were selected using the probability proportional to size sampling approach. The 250 selected enumeration areas were again stratified by sex, with 111 primary sampling units (PSUs) allocated to females and 139 PSUs to males. The differences were based on varying anticipated response rates by sex and household screening rates [25]. Splitting samples allows VACS to protect respondents' confidentiality and eliminate the chances that a perpetrator and victim of sexual violence would be interviewed [23]. In the second stage, a cluster of 25 households was selected by equal probability systematic sampling in each enumeration area. Finally, one eligible respondent, female or male, aged 13–24 years, was chosen randomly from all eligible females (or males) in each selected household. Under the Rwanda VACS design, participants aged 13–17 were considered children, and those aged 18–24 were put in the youth category. Therefore, this study's analyses focused on children, a sample of 1,110 children (618 boys and 492 girls) aged 13–17.

## Study variables

**Dependent variable.** Self-reported EV within twelve months before the survey date was the outcome variable for this study. Three types of emotional abuse were asked about in the survey: being treated unequally compared to other children in the household/family, being ignored intentionally, and being insulted repeatedly. Respondents who reported having experienced EV were also asked if that had happened to them within the last 12 months before the survey date. Based on types of EV and the time at which reported EV had happened, an outcome variable called "Emotional Violence" was constructed for this study. It was defined as having experienced at least one of the three types of EV assessed by the survey in Rwanda. In Table 1, we presented each type of EV and a question asked respondents about it.

**Independent variables.** The choice of independent variables presented in this study was informed by the WHO and CDC's classification of risk and protective factors for violence against children. According to them, risk and protective factors for violence against children have been classified into three categories: individual factors, family factors and community factors [26, 27].

Regarding individual factors, the assessed independent variables in this study comprised the child's gender, orphanhood and schooling status. For family factors, we assessed the relationships with parents, including living with parents, closeness with parents; household characteristics, including age and gender of the household head, size and wealth index, and household cover by health insurance. Finally, in community factors, we assessed the number of friends, trusting people and feeling safe.

**Table 1. Types of EV and questions that were asked of respondents for each type of EV.**

| Type of EV | Questions asked to respondents |
|---|---|
| **Unequal treatment** | • Has a parent, adult caregiver, or another adult relative ever treated you unequally compared to other children in your household/family? |
| **Intentional ignorance** | • Has a parent, adult caregiver, or another adult relative ever Ignored you intentionally? |
| **Repeated insults** | • Has a parent, adult caregiver, or another adult relative ever Insulted you repeatedly? |

For relationships with biological parents, respondents were asked if they lived with their biological mothers and fathers. They were also asked how close they felt to their biological mother and father. From the first two questions, whose response options were "live with biological mother/biological father" or "Does not live with biological mother/biological father", a new variable was generated, with three levels: living with both biological parents, living with neither biological parent and living with a single biological parent. The question about closeness with biological mother/biological father had five response options: very close, close, not close, no relationship, and don't know/ declined. We recorded the last three levels into one level: "not close/no relationship".

Regarding the child's household characteristics, the following variables were considered: the household wealth index, health insurance coverage, the number of people in the households, and the sex and age of the head of household. First, the household wealth index was generated from a household's ownership of selected assets using the Demographic and Health Survey [28] statistical procedure referred to as principal component analysis. Individual households were placed on a continuous scale of relative wealth separated into three wealth indexes: the highest wealth index, the middle health index and the lower wealth index. Third, the number of people in the household was simplified to a dichotomous variable: households with 1–4 people and households with five or more. The basis for that recording was that the mean size of a Rwandan household was four persons [29]. Finally, the head of household age was recoded into two levels: younger heads (less than 30 years) and older heads of household (31 years and more).

The last category of independent variables assessed is the child's social connectedness within the community. Respondents were asked how many friends they had, how much they talked to friends, and how much they trusted people in the community. The variable on the number of friends was recorded with two response levels: "zero friends" and "one friend and more". The question about talking to friends had five response levels (a lot, a little, not very much, not at all, and don't know/decline), and its last three groups were recoded into one level: "Does not talk to friends". In the same way, the question on trusting people in the community had five response levels (a lot, somewhat, not too much, not at all, and don't know/ decline), and its last three levels were recorded into one level: "Does not trust".

## Statistical analyses

Data analysis was performed in Stata 14.2. All statistics (descriptive statistics and logistic regression models) were adjusted using standard weighting procedures to correct for unequal probability of selection and change for non-response and to produce national results representative of the national population of children aged 13–17 years in Rwanda. The weighting procedure was applied in two steps. In the first step, a base weight for each sample respondent was performed. In the second step, base weights for nonresponse were adjusted for [25]. As a result, we reported weighted percentages with confidence intervals (CI) by gender in descriptive statistics results. Regarding the multivariable analyses, we analysed the total sample of all children, and a backward stepwise regression approach was manually applied to drop non-significant variables (p-value $\leq$ 0.05). As a result, we reported full and reduced logistic regression models.

## Ethical considerations

VACS applies important considerations adhering to the World Health Organization's recommendations on ethics and safety in violence studies to ensure the ethical protection of children and young people who participate in the survey [30, 31]. Rwanda VACS considered, adapted,

and pre-tested VACS standard tools before full implementation in the country. A NATIONAL TECHNICAL WORKING GROUP DEVELOPED Rwanda VACS protocol and tools, with technical support from CDC and UNICEF-Rwanda. The Rwanda National Ethics Committee (RNEC) and the CDC-Institutional Review Board independently reviewed and approved the study. Questionnaires and other survey tools were translated into Kinyarwanda and pre-tested before use. Trained male interviewers conducted interviews with male respondents and trained female interviewers with female respondents, and respondents consented to participate in the survey.

There was a two-step process to obtain consent. The first included obtaining the permission of the head of household or any other adult acting as the head of household to survey the selected household and talk to the selected respondent. After the head of household/adult had agreed to the survey to be conducted in the household, he was asked to participate in the household questionnaire, and the interviewer conducted the head of the household interview. The second step consisted of obtaining the consent of child/youth respondents. Before interviews, after permission to complete the survey in the selected household and to speak to the selected respondents was obtained from the head of the household, informed consent was also obtained from each selected respondent. The interviewer received informed assent in households where the respondent chosen was a minor (13–17 years old). A similar consent process was followed where the respondent was an emancipated minor (18–24 years old) or lived in a child-headed household, except that parental/caregiver permission was not necessary [25].

## Results

### Prevalence of reported EV

Nine percent (8.87%, 95% CI [6.95–11.25]) of male children and five percent (5.17%, 95% CI [3.79–7.03]) of female children reported having experienced EV in their lifetime. Seven percent (6.77%, 95% CI [5.15–8.84]) of male children and four percent of female children (3.97%, 95% CI [2.83–5.54]) reported having experienced EV in the last twelve months before the survey. Fathers and mothers were the top two perpetrators of EV against children. Seventeen percent of male children (17.09%, 95% CI [11.06–25.47]) and twelve percent of female children (11.89%, 95% CI [6.97,19.55]) experienced EV by their fathers. Mothers were responsible for nineteen percent (19.25%, 95% CI [11.86–24.42]) of EV reported by male children and eleven percent (10.78%, 95% CI [5.77–19.25]) of EV reported by female children. More details on the prevalence of EV reported by children are summarized in Table 2.

### Characteristics of children who reported EV

One in four male children (25.53%, 95% CI [17.41–35.79]) and approximately two female children (18.99%, 95% CI [12.28–28.19]) who experienced EV were living with both of their biological parents. Approximately two male children (18.90%, 95% CI [11.9–28.68]) and six percent of female children (5.96%, 95% CI [2.89–11.88]) who reported EV lived with neither of their biological parents and another nineteen percent of male children (18.99%, 95% CI [12.28–28.19]) and twelve percent of female children who reported EV lived with a single biological parent (mother or father). Regarding reasons for not living with biological parents in children who reported EV and indicated that they were not living with biological parents during the survey, seventeen percent (17.17%, [8.691–31.14]) of male children and fourteen percent of female children (13.88%, 95% CI [6.305–27.84]) who reported EV indicated that deaths of parents for not living with them. In addition, parents' remarriage/divorce was the cause for not living with biological parents in approximately one in four male children (26.23%, 95% CI [14.61–42.5]) and in seven percent of female children (7.01%, 95% CI [2.795–16.51]) who

**Table 2. Prevalence and perpetrators of EV reported by children by gender—Rwanda violence against children and youth survey, 2015.**

| Prevalence and perpetrators | Male children | | Female children | |
|---|---|---|---|---|
| **Prevalence of EV (Male: N = 618 Female: N = 492 Total: N = 1110-)** | **%¥** | **[95% CI]** | **%¥** | **[95% CI]** |
| Emotional violence (Lifetime) | 8.87 | [6.95–11.25] | 5.17 | [3.79–7.03] |
| Emotional violence (Last 12 months) | 6.77 | [5.15–8.84] | 3.97 | [2.83–5.54] |
| **Incidence of EV (among children who reported EV in the last 12 months: Male n = 78- Female n = 42- Total n = 120)** | | | | |
| Once | 2.75 | [0.85–8.56] | 0.00 | 0.00 |
| Few | 32.50 | [22.94–43.78] | 16.27 | [10.31–24.73] |
| Many | 27.76 | [19.21–38.31] | 20.72 | [13.04–31.3] |
| **Perpetrators of EV (among children who reported EV in the last 12 months: Male n = 78, Female n = 42, Total n = 120)** | | | | |
| Child father | 17.09 | [11.06–25.47] | 11.89 | [6.97,19.55] |
| Child mother | 19.25 | [12.94–27.65] | 10.78 | [5.77–19.25] |
| Child brother | 1.77 | [0.42–7.14] | 0.00 | 0.00 |
| Child sister | 0.83 | [0.11–5.77] | 0.51 | [.07–3.68] |
| Child aunt/uncle | 8.85 | [4.32–17.29] | 3.04 | [1.01–8.74] |
| Child stepfather | 0.84 | [0.11–5.85] | 2.58 | [0.58–10.76] |
| Child stepmother | 6.22 | [2.00–17.75] | 6.14 | [2.78–13.02] |
| Others (community members) | 7.86 | [4.39–13.68] | 2.37 | [0.83–6.60] |

**Note**

CI = Confidence Interval

¥Nationally representative weighted percentages

reported EV. Fifteen percent of male children (14.93%, 95% CI [7.13–28.64]) and eight percent of female children (8.36%, 95% CI [3.483–18.74]) were not living with their biological parents because the latter abandoned them.

Male children who experienced EV had a closer relationship with their mothers than their fathers. Over one in three male children (33.55%, 95% CI [23.84–44.89]) reported feeling very close to their biological mothers, and only approximately nineteen percent (18.33%) of them reported feeling very close to their biological fathers. A similar trend was observed in female children who reported EV. Over three in ten male children (34.81%, 95% CI [26.51–44.15]) and one in four female children (25.10%, 95% CI [16.32–36.54]) who experienced EV in two children (51.1%) who experienced EV did not feel close to their biological parents.

Slightly four male children (45.15%, 95% CI [35.07–55.65]) and three of three female children (30.97%, 95% CI [21.88–41.82]) who experienced EV lived in larger households with more than five people. Slightly over two in ten male children (22.06%, 95% CI [14.43–32.19]) and over one in ten female children (15.47%, 95% CI [9.497–24.18]) who reported EV belonged to households from the lowest wealth quintile. Additional details on the characteristics of children who reported emotional violence are presented in Table 3.

## Correlates of EV among children in Rwanda

The results on correlates of EV against children are presented in Table 4. Four categories of correlates were investigated for association with EV: the child's characteristics, the child's relationship with biological parents, household characteristics, and the child's community social

**Table 3. Characteristics of children who reported EV, by gender—Rwanda violence against children and youth survey, 2015.**

| Characteristics | Male | | Female | |
|---|---|---|---|---|
| | n = 78 | | n = 42 | |
| | %¥ | [95% CI] | %¥ | [95% CI] |
| **Orphan hood** | | | | |
| Not an orphan | 44.2 | [33.79–55.14] | 27.86 | [19.36–38.31] |
| Orphan one parent | 19.23 | [11.49–30.41] | 8.71 | [4.292–16.88] |
| **Schooling status** | | | | |
| Attend school | 41.05 | [31.3–51.56] | 22.59 | [14.33–33.73] |
| Does not attend school | 21.96 | [13.98–32.76] | 14.4 | [8.731–22.84] |
| **Living with parents** | | | | |
| Live with both parents | 25.53 | [17.41–35.79] | 18.99 | [12.28–28.19] |
| Live with neither parent | 18.9 | [11.9–28.68] | 5.96 | [2.894–11.88] |
| Live with a single parent (mother or father) | 18.58 | [12.73–26.31] | 12.04 | [6.286–21.82] |
| **Reasons for not living with parents (among children who reported not living with at least one biological parent on the survey date)** ⊆ | | | | |
| Death | 17.18 | [8.691–31.14] | 13.88 | [6.305–27.84] |
| Work-Studies | 9.79 | [4.588–19.68] | 0.89 | [.1156–6.516] |
| Parents remarried/divorced | 26.23 | [14.61–42.5] | 7.01 | [2.795–16.51] |
| Abandoned by parents/other reasons | 14.93 | [7.13–28.64] | 8.36 | [3.483–18.74] |
| **Closeness with mother** | | | | |
| Very close with mother | 33.55 | [23.84–44.89] | 15.5 | [9.288–24.74] |
| Close with mother | 18.27 | [11.48–27.82] | 5.34 | [2.496–11.06] |
| Not close with mother | 11.18 | [6.468–18.65] | 16.15 | [9.196–26.8] |
| **Closeness with father** | | | | |
| Very close with father | 18.33 | [12.27–26.49] | 12.5 | [6.704–22.12] |
| Close with father | 12.79 | [6.695–23.06] | 5.31 | [2.064–12.98] |
| Not close with father | 31.89 | [23.77–41.28] | 19.18 | [12.29–28.68] |
| **Closeness with biological parents** | | | | |
| Very close with biological parents | 9.45 | [5.149–16.71] | 7.18 | [3.124–15.65] |
| Close with biological parents | 18.75 | [11.88–28.33] | 4.71 | [1.718–12.26] |
| Not close with biological parents | 34.81 | [26.51–44.15] | 25.1 | [16.32–36.54] |
| **Gender of the head of household** | | | | |

(*Continued*)

**Table 3.** (Continued)

| Characteristics | Male | | Female | |
|---|---|---|---|---|
| | **n = 78** | | **n = 42** | |
| | **%¥** | **[95% CI]** | **%¥** | **[95% CI]** |
| Male | 40.28 | [30.51–50.9] | 29.68 | [20.74–40.49] |
| Female | 22.73 | [14.72–33.38] | 7.31 | [3.28–15.51] |
| **Age of head of household** | | | | |
| Less than 30 years | 3.19 | [1.202–8.167] | 2.99 | [1.084–7.988] |
| 31 years and more | 59.83 | [48.37–70.3] | 34 | [23.93–45.75] |
| **Number of people in the household** | | | | |
| 1–4 people | 17.86 | [11.83–26.05] | 6.02 | [2.491–13.82] |
| Five people and more | 45.15 | [35.07–55.65] | 30.97 | [21.88–41.82] |
| **Household wealth index** | | | | |
| Higher wealth index | 23.54 | [15.72–33.71] | 9.54 | [4.395–19.48] |
| Middle wealth index | 17.41 | [11.52–25.44] | 11.98 | [7.166–19.37] |
| Lower wealth index | 22.06 | [14.43–32.19] | 15.47 | [9.497–24.18] |
| **Household covered by health insurance**. | | | | |
| Covered by a health insurance | 48 | [38.37–57.79] | 25.04 | [16.82–35.56] |
| Not Covered by a health insurance | 15.01 | [9.627–22.64] | 11.95 | [6.708–20.39] |
| **Number of friends** | | | | |
| One friend and more | 55.7 | [44.57–66.29] | 29.94 | [20.91–40.85] |
| No friend | 7.31 | [3.592–14.3] | 7.05 | [3.335–14.31] |
| **Trusting people in the community** | | | | |
| Trust people a lot | 21.39 | [14.27–30.79] | 10.49 | [5.627–18.73] |
| Trust people somewhat | 17.92 | [12.05–25.81] | 6.05 | [2.879–12.27] |
| Does not trust people | 23.7 | [15.9–33.79] | 20.45 | [13.06–30.55] |
| **Feeling safe in the community** | | | | |
| Feels very safe | 28.71 | [21–37.88] | 13.23 | [7.85–21.44] |
| Feels somewhat safe | 31.98 | [23.61–41.71] | 15.2 | [9.158–24.16] |
| Does not feel safe | 2.32 | [.6998–7.4] | 8.56 | [4.276–16.4] |

**Note**:

**CI** = Confidence Interval

¥Nationally representative weighted percentages

⊆ The sample size for this variable comprises 48 male children and 21 female children who experienced EV and indicated that they did not live with at least one of their biological parents

**Table 4. Correlates of EV against children—Rwanda violence against children and youth survey, 2015.**

| Correlates | Full Model of reported EV in all children (n = 1073) | | | Reduced Model of reported EV in all children (n = 1073) | | |
|---|---|---|---|---|---|---|
| | OR | [95% CI] | p-value | OR | [95% CI] | p-value |
| **Child sex** | | | | | | |
| Male | Ref. | | | Ref. | | |
| Female | **0.47** | **[0.29–0.74]** | **0.001** | **0.48** | **[0.31–0.76]** | **0.002** |
| **Schooling status** | | | | | | |
| Attend school | Ref. | | | Ref. | | |
| Does not attend school | **1.79** | **[1.09–2.92]** | **0.021** | **1.80** | **[1.11–2.92]** | **0.018** |
| **Orphan hood** | | | | | | |
| Not an orphan | Ref. | | | | | |
| Orphan | 1.43 | [0.72–2.83] | 0.302 | | | |
| **Live with both parents.** | Ref. | | | Ref. | | |
| Live with neither parent | 1.61 | [0.75–3.46] | 0.225 | 1.54 | [0.72–3.33] | 0.267 |
| Live with mother only | 1.17 | [0.48–2.87] | 0.728 | 0.86 | [0.43–1.72] | 0.666 |
| Live with father only | **2.96** | **[1.15–7.67]** | **0.025** | **3.08** | **[1.21–7.85]** | **0.019** |
| **Closeness with mother** | | | | | | |
| Very close with mother | Ref. | | | | | |
| Close with mother | 1.08 | [0.58–2.02] | 0.814 | | | |
| Not close with mother | 1.65 | [0.82–3.30] | 0.16 | | | |
| **Closeness with father** | | | | | | |
| Very close with father | Ref. | | | Ref. | | |
| Close with father | 0.41 | [0.16–1.04] | 0.06 | **0.39** | **[0.15–0.99]** | **0.048** |
| Not close with father | 0.70 | [0.21–2.37] | 0.57 | 0.52 | [0.17–1.63] | 0.264 |
| **Closeness with biological parents** | | | | | | |
| Very close with biological parents | Ref. | | | Ref. | | |
| Close with biological parents | **3.26** | **[1.12–9.44]** | **0.030** | **3.46** | **[1.31–9.14]** | **0.012** |
| Not close with biological parents | **4.59** | **[1.21–17.38]** | **0.025** | **7.18** | **[2.12–24.37]** | **0.002** |
| **Gender of the head of household** | | | | | | |
| Male | Ref. | | | | | |
| Female | 0.71 | [0.32–1.55] | 0.384 | | | |
| **Number of people in the household** | | | | | | |
| 1–4 people | Ref. | | | Ref. | | |
| Five people and more | 1.73 | [0.94–3.19] | 0.08 | **1.81** | **[1.03–3.19]** | **0.039** |
| **Household wealth index** | | | | | | |
| Higher wealth index | Ref | | | Ref. | | |
| Middle wealth index | **0.56** | **[0.33–0.97]** | **0.038** | **0.57** | **[0.34–0.97]** | **0.038** |
| Lower wealth index | **0.56** | **[0.32–0.99]** | **0.048** | 0.60 | [0.34–1.04] | 0.07 |
| **Household health insurance** | | | | | | |
| Covered by a health insurance | Ref. | | | | | |
| Not Covered by a health insurance | 1.24 | [0.79–1.93] | 0.35 | | | |
| **Number of friends** | | | | | | |
| One friend and more | Ref | | | Ref. | | |
| No friend | **2.03** | **[1.03–4.01]** | **0.041** | **2.08** | **[1.05–4.11]** | **0.035** |
| **Trusting people in the community** | | | | | | |
| Trust people a lot | Ref | | | Ref. | | |
| Trust people somewhat | **0.47** | **[0.24–0.94]** | **0.031** | **0.47** | **[0.23–0.93]** | **0.029** |
| Does not trust people | 1.29 | [0.63–2.64] | 0.49 | 1.29 | [0.64–2.60] | 0.467 |

*(Continued)*

**Table 4.** (Continued)

| Correlates | Full Model of reported EV in all children (n = 1073) | | | Reduced Model of reported EV in all children (n = 1073) | | |
|---|---|---|---|---|---|---|
| | OR | [95% CI] | p-value | OR | [95% CI] | p-value |
| Feeling safe in the community | | | | | | |
| Feels very safe | Ref. | | | Ref. | | |
| Feels somewhat safe | 1.02 | [0.57–1.82] | 0.945 | 1.03 | [0.58–1.82] | 0.927 |
| Does not feel safe | 2.52 | [0.99–6.47] | 0.054 | **2.56** | **[1.03–6.39]** | **0.044** |

**Note**:

**OR**: odds ratio

**Ref.**: Reference

**CI** = Confidence Interval

connectedness. For child characteristics, female children were less likely to report EV than male children (OR = 0.48, 95% CI = [0.31–0.76]). In addition, the odds of reporting EV were more significant among children who were not going to school (OR: 1.80, 95% CI: [1.11–2.92]) than among those who were going to school.

Regarding child relationship with biological parents, children living with their fathers only (OR = 3.08, 95% CI = [1.21–7.85]) and those who reported not feeling close to biological parents (OR = 7.18, 95% CI = [2.12–24.37]) had greater odds of reporting EV than children who lived with both biological parents and those who felt very close to their birth parents.

In the child's household characteristics, the likelihood of experiencing EV decreased among children from the middle wealth index households (OR = 0.57, 95% CI = [0.34–0.97]) than for children from the highest wealth index households. Additionally, children from larger households of five people and more were more likely to experience EV (OR = 1.81, 95% CI = [1.03–3.19]) than children in smaller households comprising one to four people. Last, regarding child community social connectedness, we found that not having a friend (OR = 2.08, 95% CI = [1.05–4.11]) and not feeling safe in the community (OR = 2.56, 95% CI = [1.03–6.39]) were positively associated with EV against children.

## Discussion

This study had two objectives: describing patterns of EV against children and the profile of affected children and assessing factors associated with EV against children in Rwanda.

On the first objective, we found that one in ten children had experienced EV by a parent, caregiver, or adult relative within twelve months before the survey and that fourteen percent had been emotionally abused by a parent, caregiver, or adult relative in their lifetime. One in two children emotionally abused had experienced many incidences of EV. The prevalence of EV against children found in Rwanda was in the same range as the prevalence of EV reported by children in similar surveys in Botswana [32] and Kenya [33]. However, it was below the prevalence of EV reported by children in another similar study in Uganda [17]. Parents topped the list of perpetrators of EV against children. Fathers, mothers, stepfathers and stepmothers were responsible for approximately eighty percent of all incidences of EV reported by children. This finding is consistent with global literature and estimates of violence against children, which indicated that parents were the most common perpetrators of EV against children, with prevalence rates exceeding 50% in some instances [18, 34, 35].

In analyses conducted on the profile of children who reported EV, we found that more than one in three emotionally abused children were not living with their birth parents because of

parental remarriage or divorce, and over one in five children who reported EV were not living with their birth parents because of parental abandonment. However, there is still a debate on the negative and positive effects of not living with biological parents on child outcomes. Some researchers who studied this issue concluded that these effects vary across societies and are mediated by the quality of the relationship between parents and how they relate with their children [36–44]. Nonetheless, parental separation and divorce are painful processes that leave children disarmed, prevent them from enjoying their parental love to the fullest, and potentially harm the child's psychosocial development [45, 46]. Additionally, underlying causes of parental remarriage and divorce remain major family stressors [40, 43, 47, 48] that deprive children of a conducive family environment to thrive and make them vulnerable to all sorts of child maltreatment.

For children who were not living with their birth parents because of parental abandonment, it should be noted that it is increasingly recognised as a form of child maltreatment and neglect, and it may constitute a crime in some situations [49, 50]. However, many reasons would push parents to abandon a child. Examples include poverty and financial hardship, family violence, unwanted pregnancy, lack of social support, etc. [50, 51] are some known factors behind child abandonment. All these constitute underlying factors that expose children to childhood adversity, including emotional abuse. In addition, children have more expectations from their parents, and parental abandonment creates low self-esteem in abandoned children. Therefore, more focused studies on children not living with their biological parents for different reasons mentioned by children in the Rwanda survey should be conducted to better understand and document their outcomes and quality of life.

The second objective of this study was to assess factors associated with EV against children. The factors considered by this study were in three categories: child, family, and community factors.

Regarding the child's factors, we found that being a female child was negatively associated with the risk of EV. Children out of school had more odds of being emotionally abused than children attending school. The gender difference in the occurrence of EV in children in Rwanda was consistent with the findings of a 2017 Global Report on Ending Violence in Childhood. Data in this report indicated that, similar to physical violence against children, emotional violence against children was commonly experienced within the home and was mainly perpetrated against boys by relatives and neighbours. On girls, the report found that emotional violence tended to be perpetrated by partners outside the home [52]. It is also worth noting that the Rwanda violence against children and youth survey asked only about emotional violence happening around the home (see Table 1), which is consistent with the 2017 Global Report on Ending Violence in Childhood and could explain why female children were less likely to report emotional violence than male children.

[53, 54] For children out of school, according to Rwanda's primary and secondary education system, children aged 13–17 are supposed to be pursuing primary and secondary education. However, given that the Government of Rwanda promotes compulsory school fee-free education for all children, the fact that children out of school were more at risk of EV than children in school would be an indication that these children come from disadvantaged family settings that are unable to provide them with necessary and additional socioeconomic support to pursue education [55].

This context of an unsupportive family environment also allowed us to interpret family factors found to be associated with EV. Children living with their fathers only and those not feeling close to their birth parents had more risks of experiencing emotional violence than their fellows who lived with both parents and felt close to their biological parents, respectively. However, feeling close to the birth father was a protective factor against EV. The finding that lack of

parent-child closeness increased the risk of EV for children should be interpreted taking into account that a significant number of children who were emotionally abused in Rwanda did not live with their birth parents because of their parent's death, remarriage, divorce or abandonment. Studies have demonstrated that such family structure disruptions impact children's well-being and psychosocial development [56].

The quality of the child-parent relationship may also imply the socioeconomic status of the households in which children live [57]. Our findings indicated that children in larger families of five people and more had more risks of being emotionally abused than children in smaller families, while living in households in the middle wealth index was a protective factor against EV. Many researchers have previously stated that larger households presented a higher risk of child abuse. However, an increasing number of studies question those findings by showing that household size should not be measured as a unilateral risk factor [58]. Instead, it must be interpreted together with other family parameters and backgrounds. Thus, in the Rwanda survey, we observed that half of the emotionally abused children were not living with their birth parents, meaning they lived within foster families. As nine out of ten children were not separated from their parents because of deaths, remarriage, divorce, or abandonment, the current living of these children and how they relate with their foster families may be influenced by the process and events that caused separation from their birth parents.

Last, regarding community factors, while a large part of emotional violence against children seemed to be committed in the family circles in Rwanda, children who reported not trusting people in the community, children who did not have friends and those who did not feel safe in their community were vulnerable to emotional abuse.

## Limitations

This study is the first to analyze nationally representative data on emotional violence in Rwanda and present the profile of children affected by emotional violence and its associated factors. However, it has some limitations related to its design, which should be considered when interpreting its findings. First, data collected by this study are self-reported, and biased responses might have been provided due to a misunderstanding of what was asked or the social-desirability bias for respondents who would have wanted to look good [59]. Second, as a cross-sectional study, it was impossible to determine direct causal associations.

## Conclusion

As per this study's findings, emotional violence against children is pervasive in Rwanda, and parents remain critical actors. Being a female child, being close to the birth father, residing in a household from the middle wealth index, and trusting people in the community emerged as protective factors against EV in Rwanda. Conversely, children not attending school, children living with their fathers only, children without a close relationship with biological parents, children from larger households, children without a friend, and children who reported not feeling safe in their communities were identified as groups of children vulnerable to emotional violence in Rwanda.

A family-centred approach, focusing on positive parenting and protecting vulnerable children, is needed to reduce emotional abuse and associated factors. In addition, there should be more follow-up studies and systematic data collection on these categories of children to know more about their experiences of EV and inform the surveillance of child maltreatment in Rwanda.

## Author Contributions

**Conceptualization:** Alypio Nyandwi, Cyprien Munyanshongore, Laetitia Nyirazinyoye, Prata Ndola.

**Data curation:** Alypio Nyandwi.

**Formal analysis:** Alypio Nyandwi.

**Methodology:** Alypio Nyandwi.

**Supervision:** Cyprien Munyanshongore, Laetitia Nyirazinyoye.

**Writing – original draft:** Alypio Nyandwi.

**Writing – review & editing:** Alypio Nyandwi, Cyprien Munyanshongore, Laetitia Nyirazinyoye, Prata Ndola, Gisela Perren-Klingler.

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
