## [Decision Letter · Decision Letter 0]

19 Oct 2022

PONE-D-22-24606Correlates of Emotional Violence against Children in RwandaPLOS ONE

Dear Dr. Nyandwi,

Thank you for submitting your manuscript to PLOS ONE. After careful consideration, we feel that it has merit but does not fully meet PLOS ONE’s publication criteria as it currently stands. Therefore, we invite you to submit a revised version of the manuscript that addresses the points raised during the review process.

We look forward to receiving your revised manuscript.

Kind regards,

Ari Samaranayaka, PhD

Academic Editor

PLOS ONE

Journal Requirements:

3. We note you have included a table to which you do not refer in the text of your manuscript. Please ensure that you refer to Tables 1,2 and 3 in your text; if accepted, production will need this reference to link the reader to the Table.

**Additional Editor Comments:**

Aim: This study used data from the Rwanda Survey to examine patterns of emotional violence (EV) and associated, and to describe the profile of children who were emotional abused.

Says “All data are within the manuscript and supporting documents”. But no supporting documents found.

Agree with authors view that “identification of emotional violence largely depends on the dialogue with the victim who, through his narrated experiences, can disclose about his experiences of emotional violence. Consequently, emotional violence tends to be hidden and might be underestimated or given less attention than child physical and sexual abuse”. Readers need at least briefly know how these were tackled in methods, even if they were detailed in previous publications. Specifically, what were the measures taken to mitigate possible information bias in survey data collection from children? In other words, can they remain as limitations or points to consider in future research?

Data analysis. Abstract says sampling weights were used only for descriptive statistics (not for logistic regression). But data analysis paragraph says logistic regression is weighted for the complex sampling design. Which of these was used, and why? Need to be clear when the sampling weights were used, and reasons for not using them for other procedures (if any). Some of the related older literature are: (A) Gary Solon, Steven J. Haider, Jeffrey M. Wooldridge Solon (2015). What Are We Weighting For? The journal of human resources 50(2): 301-316. (B) Christopher Winship and Larry Radbill (1994). Sampling weights and regression analysis. sociological methods & research. 23(2):230-257.

Table3 refers to a full model and a reduced model. I assume full model is the multivariable model that included all the variables identified from unreported individual univariable logistic models (Identified as Pvalue less than or equal to 0.1). Why OR from this multivariable model is labelled as uOR? What is reduced model? Data analysis section has no mention of such a model.

I have some confusion with Table1 contents. I assume it has a formatting issue. Contents in subheadings rows are not aligned with columns. So I assume the first column is for males, second column is for females, and 3rd column is for total. If so why percentages for the total is larger than those for males and females? Same issue in table2 too. Also, I noted the percentages for Perpetrators of emotional violence for total sample column sum to 100. Does that mean no one had multiple perpetrators?

Miner comments.

Spellings: ow-income to be low income.

Spellings: “To obtained the adjusted logistic regression models,…” should be “To obtain the adjusted logistic regression models, …”.

Older heads of household is 31+ years (not 51+ years)?

Some subheading numbers in the manuscript not matching to their main headings

Please remove table3 footnote and associated asterisks, they are redundant.

Reviewers' comments:

Reviewer's Responses to Questions

**Comments to the Author**

1. Is the manuscript technically sound, and do the data support the conclusions?

Reviewer #1: Yes

Reviewer #2: Partly

2. Has the statistical analysis been performed appropriately and rigorously? 

Reviewer #1: No

Reviewer #2: Yes

3. Have the authors made all data underlying the findings in their manuscript fully available?

Reviewer #1: No

Reviewer #2: Yes

4. Is the manuscript presented in an intelligible fashion and written in standard English?

Reviewer #1: No

Reviewer #2: Yes

5. Review Comments to the Author

Reviewer #1: Reviewer comments

Manuscript Number: PONE-D-22-24606

Title "Correlates of Emotional Violence against Children in Rwanda".

Generally speaking:

Thank you for providing me the opportunity to review this manuscript that raises important and interesting issues about correlates of emotional violence against children in one of the developing countries.

Comment 1:

1. TITLE:

The title is not scientifically written. It needs to be revised.

Comment 2:

2. ABSTRACT:

a) The aim needs to be revised. It is not scientifically written.

b) Methodology has to mention the elements included as measuring parameter but without details.

c) No need for details of methodology in the abstract. Only main points. It should not include the type of statistical analysis.

d) Methodology is lacking the study design.

Comment 3:

3. INTRODUCTION:

a) The text of the introduction does not seem to have coherence and integrity. Content is scattered. It should be concise and targeted to the aim.

b) Newest Global/ Regional/ Rwanda prevalence of violence against children should be stated.

c) The introduction lacks literature on emotional violence against children.

d) Risk factors associated with emotional violence against children should be clearly stated.

e) The current situation of other developing and developed countries should also be added.

f) Explaining why this topic was chosen for analysis in this article is not well written. The benefits of conducting the study to the community should be explained.

g) The aim should be clearly stated.

Comment 4:

4. METHODS:

a) The methods will be better if structured in this way: type of the study, study setting, study participants, study tools, sample size, data analysis, and ethical considerations.

b) Type of the study should be added.

c) Sample size should be mentioned.

d) The characteristics of the study participants should be mentioned as inclusion criteria and exclusion criteria (if any)

e) The head of household age was recoded into two levels: younger heads of household (less than 30 years) and older heads of household (51 years and more), what about the heads of household aged 30-51 years.

f) Were the samples normal or not?

g) There was no clear mention of the questionnaire(s) used.

h) What literature was reviewed to develop the questionnaire? The authors do not provide any references. Was it piloted to assess its internal consistencies? Was it validated? Additional explanations about the validity and reliability of the questionnaire should be added - explained step by step. (Content and structure validity).

i) It is advisable to include the questions as per each domain in the methodology. How to score the questionnaire, specify the cut points in the questionnaire?

j) How long did it take to complete each questionnaire? Mention the number of questions in the questionnaire. Did the participant answer it themselves or with assistance from researcher?

k) What strategy was devised to increase the accuracy of the study and the accuracy of the answers? What types of technique the authors used to keep/control the data quality?

l) Authors should include a reference for using the stated formula in calculating the sample size. Furthermore, the basis of sample size calculation should be mentioned to know the confidence level and the margin of error. How did you determine the sample size? Please give a justification for the sample size.

m) How did the authors measure their outcome variables? What types of data (quantitative and/or qualitative) were used?

n) Tests of significance for each type of data should be added. Clearly state for which variables each test was performed.

o) P-values of less than 0.05 are the usual threshold used to determine significance.

p) Please describe more clearly the variable selection. Which variables were considered for univariate analysis? Which variables were included for further modelling based on the univariate analysis? Which kind of modelling was used for the further analysis?

q) How did the authors describe household wealth index as highest, middle, and the lower wealth indexes. Could they specify the cutoff points for that.

r) Ethical considerations: informed consent form and confidentiality in this study need to be mentioned in this section too.

Comment 5:

5. RESULTS:

a) In the result section, it is advisable to add test of significant in each table from table 1 and 2.

b) Please provide more information about the regression (selection process, selected models). Variables entered in the logistic regression should be mentioned.

c) Age was not included in the final regression model; age might contribute to EV.

d) No need to conduct univariate logistic regression because it doesn’t consider the confounding variables.

e) What was the criteria to pass your variables from univariate to multivariate logistic regression?

f) Titles of the tables has to be improved to be more specific.

g) Any abbreviations should be mentioned below the tables as CI, aOR, and uOR in table 3.

h) It is advisable to combine tables 1 and 2 in one table.

i) In table 1: the name of each column header is incomplete.

Comment 6:

6. DISCUSSION:

a) It is advisable to explain the study objective at the beginning of the discussion.

b) Please start the discussion with a short summary of the study and the main findings.

c) There were no explanations for prevalence rates in different area. The manuscript could be greatly strengthened if the authors could provide highlight on the prevalence of EV against children in other developing and developed countries with similar context.

d) The discussion is very short, and the authors have not done a good job of connecting their findings to the literature and interpreting their findings. Discuss by using the scientific reasoning the predictors/ correlates of EV against children in other developing and developed countries with similar context. Compare the findings of the study with other findings and state the reasons for the strengths and weaknesses in each section.

e) The discussion can be further strengthened by adding more relevant and recent references.

Comment 7:

7. CONCLUSION:

Mention factors separately (positively and negatively associated).

Comment 8:

8. STRENGTHS AND LIMITATIONS:

a) Please analyze the strengths of the study.

b) How the limitations affect the generalizability of the findings.

Comment 9:

9. REFERENCES:

Please revise reference no. 30

Reviewer #2: Unfortunately the research lacks originality. There is substantial amount of literature on this topic from Rwanda settings. Different population groups (age and gender) have been studied. Examples include:

Pierre NJ, Claire MM. Effect of child abuse on students’ academic performance in public secondary schools in Rwanda. Journal of Education. 2021 May 28;4(2).

Blanchette I, Rutembesa E, Habimana E, Caparos S. Long-term cognitive correlates of exposure to trauma: Evidence from Rwanda. Psychological trauma: theory, research, practice, and policy. 2019 Feb;11(2):147.

Bahati C, Izabayo J, Munezero P, Niyonsenga J, Mutesa L. Trends and correlates of intimate partner violence (IPV) victimization in Rwanda: results from the 2015 and 2020 Rwanda Demographic Health Survey (RDHS 2015 and 2020). BMC women's health. 2022 Dec;22(1):1-3.

Thomson DR, Bah AB, Rubanzana WG, Mutesa L. Correlates of intimate partner violence against women during a time of rapid social transition in Rwanda: analysis of the 2005 and 2010 demographic and health surveys. BMC women's health. 2015 Dec;15(1):1-3.

The manuscript adds very little to existing knowledge base in this area.

6. PLOS authors have the option to publish the peer review history of their article (what does this mean?). If published, this will include your full peer review and any attached files.

Reviewer #1: No

Reviewer #2: No

---

## [Author Response · Author response to Decision Letter 0]

18 Nov 2022

I am very thankful to the reviewers who have taken their time and put their efforts into reviewing my manuscripts. I have no doubt that their comments have been very critical in improving the quality and content of my manuscript. May God Bless you all you. I have attached the a document summarizing my feedback on the reviewers and editor comments.

---

## [Decision Letter · Decision Letter 1]

12 Dec 2022

PONE-D-22-24606R1Correlates of Emotional Violence against Children in Rwanda: Findings from a Cross-Sectional National SurveyPLOS ONE

Dear Dr. Nyandwi,

Thank you for submitting your manuscript to PLOS ONE. After careful consideration, we feel that it has merit but does not fully meet PLOS ONE’s publication criteria as it currently stands. Therefore, we invite you to submit a revised version of the manuscript that addresses the points raised during the review process.

We look forward to receiving your revised manuscript.

Kind regards,

Ari Samaranayaka, PhD

Academic Editor

PLOS ONE

Additional Editor Comments:

(1) Study title has been changed, and whole manuscript fully re-written. That has largely improved the submission. However still I have below concerns.

(2) Authors reported in the introduction “… the prevalence of EV among children aged 13-17 by a parent, caregiver or adult relative in the past 12 months ranged from ten to over forty percent. The lowest prevalence was reported in Lesotho, with 6.9% of EV in female children and 3.8% of EV in male children[16]. The highest prevalence was reported in Uganda: 22.2% in girls and 22.8% in boys”. These two sentences contradictory to each other. Also the last sentence in the same paragraph is a repetition.

(3) Surveys were conducted by trained interviewers, I assume the surveys were face-to-face. Why the questionnaires and other survey tools were translated into Kinyarwanda, and back-translated into English?

(4) tables 1 and 2. I know percentages are weighted, therefore simple arithmetic not working, But, how can prevalence of EV become larger for overall (ie; all children) than for males and females across all dimensions? I asked this in the last review too.

(5) Table1. "percentage not presented due to unstable estimate". I can imagine situations that can arise this situation. But I can't see why this occur here. For example, taking incidence of EV among females, n=42. Of them if 16.3% (about 7 females) experience it few times, and 20.7% (about 9 females) experience it many times, the balance of about 26 (I said "about" because can change due to weighting for study design) should experience it only once. I can’t think of reasons for 26 out of 42 percentage estimate becomes unstable even if estimator is a weighted one.

(6) Table1. Perpetrators of EV. Percentages in 'all children' column sum to 100. Does this say none of the children had more than one type of perpetrator (ie, not counted under multiple perpetrator groups)? I assume it should be the case if it was used as a categorical variable in logistic regression (which is unclear to me from the methods or results). I asked the same question in last review too. Similarly, sums of the percentages in each of the other two columns are much lower to 100%. How can that be explained if all perpetrators belong to one of the listed groups? These type of apparent discrepancies have to be explained to strengthen the trustworthiness of the results.

(7) Please re-check for the typos. Example “Poverty and financial hardship, family violence, unwanted pregnancy, lack of social support, etc. [60,61] are some known behind child abandonment”.

(8) I noted some variables in tables 1 and 2 are not in table3 (in unadjusted form either). Eg: Perpetrators of EV variable. How you decided what variables initially included in multivariable model building?

Reviewers' comments:

Reviewer's Responses to Questions

**Comments to the Author**

1. If the authors have adequately addressed your comments raised in a previous round of review and you feel that this manuscript is now acceptable for publication, you may indicate that here to bypass the “Comments to the Author” section, enter your conflict of interest statement in the “Confidential to Editor” section, and submit your "Accept" recommendation.

Reviewer #1: All comments have been addressed

Reviewer #2: (No Response)

2. Is the manuscript technically sound, and do the data support the conclusions?

Reviewer #1: Yes

Reviewer #2: (No Response)

3. Has the statistical analysis been performed appropriately and rigorously? 

Reviewer #1: Yes

Reviewer #2: Yes

4. Have the authors made all data underlying the findings in their manuscript fully available?

Reviewer #1: Yes

Reviewer #2: Yes

5. Is the manuscript presented in an intelligible fashion and written in standard English?

Reviewer #1: Yes

Reviewer #2: Yes

6. Review Comments to the Author

Reviewer #1: Reviewer comments

Manuscript Number: PONE-D-22-24606_R1

Title "Correlates of Emotional Violence against Children in Rwanda: Findings from a Cross-Sectional National Survey".

Thank you for providing me the opportunity to re-review this manuscript that raises important issues about correlates of emotional violence against children in Rwanda: findings from a cross-sectional national survey in one of the developing countries.

It seems that all corrections were done.

Reviewer #2: Thank you for addressing the points which were previously highlighted in the review. The revised version makes a stronger case for suitability for acceptance.

7. PLOS authors have the option to publish the peer review history of their article (what does this mean?). If published, this will include your full peer review and any attached files.

Reviewer #1: No

Reviewer #2: No

---

## [Author Response · Author response to Decision Letter 1]

22 Jan 2023

I am very appreciative to reviewers for having recommended my manuscript for publication in PLOS ONE.

---

## [Editor Report · Decision Letter 2]

27 Jan 2023

PONE-D-22-24606R2Correlates of Emotional Violence against Children in Rwanda: Findings from a Cross-Sectional National SurveyPLOS ONE

Dear Dr. Nyandwi,

Thank you for submitting your manuscript to PLOS ONE. After careful consideration, we feel that it has merit but does not fully meet PLOS ONE’s publication criteria as it currently stands. Therefore, we invite you to submit a revised version of the manuscript that addresses the points raised during the review process.

We look forward to receiving your revised manuscript.

Kind regards,

Ari Samaranayaka, PhD

Academic Editor

PLOS ONE

Additional Editor Comments:

As a response to my comment in previous review, authors say, prevalence of EV become larger for overall (ie; all children) than for males and females across all dimensions because “ …. it is the total for what is observed for females and males”. I disagree because prevalence for males and females cannot be summed in this way. This is not visible now because “all children” column has been removed from tables 2 and 3 of R2 version (tables 1 and 2 in previous version).

Authors results lacks internal consistency; they are contradictory to themselves at some places. For example, authors haven’t explicitly reported how many children had experienced the outcome. According to table2, numbers are 78 males and 42 females. Using simple arithmetic (ignoring weightings for rough estimation) these become 12.6% males out of 618 and 8.5% females out of 492. Corresponding %s are reported as 8.9% for males and 5.2% for females. I assume differences could be due to weighting. But, why column percentages do not sum to near 100% whenever they should be? For example, every person who experienced EV should experience it at once or few times or more than few times. But I noted those percentages in table2 sum to 63% for males and 37% for females. Similarly, every person who experienced EV should have one or more perpetrators. However I noted perpetrators percentages in table2 sum to 62% for males and 37.9% for females. Also, number of children who experienced EV reported here differently as 97 males and 55 females. I assume missing values (and possibility of multiple responses for the survey questions) must also have played a role here; authors need to pay attention to them in the analysis.

This is related to the “unstable” estimate mentioned in the previous version which were replaced by zero in current submission. Does the zero observations in data is compatible with the rest of the data? What are the reasons for zero observations? Are they due to not having such people, or response being ‘missing’, or response being ‘not applicable”? Have authors done a proper data cleaning to assess these prior to data analysis? Please amend the article to remove above listed (and other) apparent inconsistencies within results. Impossible to accept an article with apparent contradictions.
---

## [Author Response · Author response to Decision Letter 2]

28 Jan 2023

I am thankful to all reviewers for their constructive and insightful comments and inputs

---

## [Editor Report · Decision Letter 3]

13 Feb 2023

PONE-D-22-24606R3

Correlates of Emotional Violence against Children in Rwanda: Findings from a Cross-Sectional National Survey

PLOS ONE

Dear Dr. Nyandwi,

Thank you for submitting your manuscript to PLOS ONE. After careful consideration, we have decided that your manuscript does not meet our criteria for publication and must therefore be rejected.

I am sorry that we cannot be more positive on this occasion, but hope that you appreciate the reasons for this decision.

Kind regards,

Ari Samaranayaka, PhD

Academic Editor

PLOS ONE

Additional Editor Comments:

Authors tried to respond to the issue of internal inconsistency in reporting findings. But the issue is still remains. For example, out of 618 boys included in the study, 101 experienced EV. 101/618=16.3%. But it is reported in table2 as EV prevalence of 8.87% . Similarly, of the 492 girls 55 experienced EV. 55/492=11.2%. But it is reported in table2 as 5.17% EV prevalence. For all children, 156 out of 954 experienced EV, this is reported as 14.04% EV. This last result seems correct because 156/1110=14.05%. However, authors say, prevalences in two genders were summed to get the overall prevalence (ie; 8.87+5.17=14.04) which is totally incorrect logic. This is an example only. There are numerous places in the manuscript with incompatible reporting, and incorrect calculations. Therefore I can't trust these calculations presented, nor the conclusions based on them.

- - - - -

---

## [Author Response · Author response to Decision Letter 3]

8 Mar 2023

I disagreed with the Editor. He suggested computing arithmetic calculations on weighted data I reported in the manuscript. Arithmetic calculations make sense for unweighted data.

---

## [Editor Report · Decision Letter 4]

20 Apr 2023

PONE-D-22-24606R4Correlates of Emotional Violence against Children in Rwanda: Findings from a Cross-Sectional National SurveyPLOS ONE

Dear Dr. Nyandwi,

Thank you for submitting your manuscript to PLOS ONE. After careful consideration, we feel that it has merit but does not fully meet PLOS ONE’s publication criteria as it currently stands. Therefore, we invite you to submit a revised version of the manuscript that addresses the points raised during the review process.

We look forward to receiving your revised manuscript.

Kind regards,

Pradeep Kumar, Ph.D.

Academic Editor

PLOS ONE

Journal Requirements:

Additional Editor Comments (if provided):

Comments

This study has two objectives: describing patterns of EV against children and the profile of affected children and assessing factors associated with EV against children in Rwanda. The study found that emotional violence against children is pervasive in Rwanda; and parents remain critical actors. Being a female child, being close to the birth father, residing in a household from the middle wealth index, and trusting people in the community emerged as protective factors against EV in Rwanda. Overall, I think this is a very interesting study highlighting important areas that can be targeted to reduce the EV against children in Rwanda. I feel that overall, the analysis generally supports the findings that are presented. However, the manuscript could benefit from further editing to improve overall readability. There are several typographical, and grammatical errors. Authors may reframe the sentence for better understanding or editing of the manuscript by someone who is proficient in the English language. The manuscript is not even standardized to a uniform font & font size. Additional comments are below:

1. Authors may add CI after the odds ratio in the abstract to see the significance of the results.

2. In the sentence “Mothers were responsible for seventeen percent (17.25 %, 95% CI [11.86-24.42]) of EV against male children, and nine percent (8.7 %, 95% CI [4.75-15.49]) of female children (8.7 %, 95% CI [4.75-15.49]) reported EV by their mothers.” Results are repeated. Authors should avoid the repetition of the results.

3. Authors should maintain the consistency during interpretation of results that were round figured by them.

4. I do not understand why some factors in tables have different sample sizes as the overall sample size for the current study is 1110. The authors should explain it.

5. A detailed literature review on predictors of emotional violence would give a clear picture as to why the authors considered them in the analysis.

6. Discussion on why a girl child is less emotionally abused is not justified by the present discussion. In fact, the authors seem to give a completely opposite picture of it.

7. For a better understanding of it, authors should create a variable “family with more or less son than a daughter or other combination of a different gender to get a clear picture of son preference.

---

## [Author Response · Author response to Decision Letter 4]

23 Apr 2023

I appreciate the comments and inputs I have received from all reviewers to improve the quality of my manuscript for publication in PLOS ONE. I look forward to additional steps leading to the publication of my manuscript in PLOS ONE

---

## [Editor Report · Decision Letter 5]

24 May 2023

Correlates of Emotional Violence against Children in Rwanda: Findings from a Cross-Sectional National Survey

PONE-D-22-24606R5

Dear Dr. Nyandwi,

We’re pleased to inform you that your manuscript has been judged scientifically suitable for publication and will be formally accepted for publication once it meets all outstanding technical requirements.

Kind regards,

Pradeep Kumar, Ph.D.

Academic Editor

PLOS ONE
---

## [Editor Report · Acceptance letter]

5 Jun 2023

PONE-D-22-24606R5 

Correlates of Emotional Violence against Children in Rwanda: Findings from a Cross-Sectional National Survey 

Dear Dr. Nyandwi:

I'm pleased to inform you that your manuscript has been deemed suitable for publication in PLOS ONE. Congratulations! Your manuscript is now with our production department. 

Kind regards, 

on behalf of

Dr. Pradeep Kumar 

Academic Editor

PLOS ONE